# Modulation of Allosteric Control and Evolution of Hemoglobin

**DOI:** 10.3390/biom13030572

**Published:** 2023-03-22

**Authors:** Maurizio Brunori, Adriana Erica Miele

**Affiliations:** 1Accademia Nazionale dei Lincei, via della Lungara, 00165 Rome, Italy; 2Department of Biochemical Sciences, Sapienza University of Rome, P.le Aldo Moro 5, 00185 Rome, Italy; 3Institute of Analytical Sciences, UMR 5280 ISA CNRS UCBL, Université Claude Bernard Lyon 1, 5 Rue de la Doua, 69100 Villeurbanne, France

**Keywords:** hemoglobin, allostery, trout HbI, trout HbIV, human HbA, Root effect, structure-function relationships

## Abstract

Allostery arises when a ligand-induced change in shape of a binding site of a protein is coupled to a tertiary/quaternary conformational change with a consequent modulation of functional properties. The two-state allosteric model of Monod, Wyman and Changeux [J. Mol. Biol. 1965; 12, 88–118] is an elegant and effective theory to account for protein regulation and control. Tetrameric hemoglobin (Hb), the oxygen transporter of all vertebrates, has been for decades the ideal system to test for the validity of the MWC theory. The small ligands affecting Hb’s behavior (organic phosphates, protons, bicarbonate) are produced by the red blood cell during metabolism. By binding to specific sites, these messengers make Hb sensing the environment and reacting consequently. HbI and HbIV from trout and human HbA are classical cooperative models, being similar yet different. They share many fundamental features, starting with the globin fold and the quaternary assembly, and reversible cooperative O_2_ binding. Nevertheless, they differ in ligand affinity, binding of allosteric effectors, and stability of the quaternary assembly. Here, we recollect essential functional properties and correlate them to the tertiary and quaternary structures available in the protein databank to infer on the molecular basis of the evolution of oxygen transporters.

## 1. Preamble

Volume 62A of Comparative Biochemistry and Physiology [1] contains 33 papers by the scientists who participated in the Alpha Helix expedition to the Amazon in November and December 1976. The purpose of this endeavor was the study of fish bloods and hemoglobins, taking advantage of the over 2000 different species of fish living in the Amazon basin and exploiting the rare opportunity to compare oxygen transport and respiration in both air-breathing and water-breathing fish. More than 80% of the published data were from experiments carried out on board the R.V. Alpha Helix of the Scripps Institution of Oceanography, supported by the National Science Foundation of the United States of America. The ship was well equipped for biochemistry and physiology: a cold room, refrigerated centrifuges, a good recording spectrophotometer, chromatographic facilities and more. There was even on board a Gibson–Durrum-stopped flow instrument for the study of rapid reactions, Brunori’s main interest at the time.

The introductory paper by Jeffries Wyman [2] begins with this statement: “If this expedition were to be given a name, it might well be called *A study of variations on a theme*. The Theme is hemoglobin; the variations, the differences in its properties and behavior which have been developed in the course of evolution to meet the special requirements of different animal forms”—a great inspiration and a solid theoretical foundation for the extensive systematic experimental work carried out on board the Alpha Helix.

The paper by Monod-Wyman-Changeux [3] on Allostery has been a fundamental revolution in the field of cellular control (reviewed by Cui and Karplus [4]). This elegant and powerful theory (also known as the MWC allosteric model) was a conceptual revolution in the field of biological regulation, having introduced a conformational selective mechanism to understand how enzymes cope with the ever-changing physiological demands of the organism. The population shift of conformational variants with different function has been also applied to some monomeric proteins involved in protein–protein recognition events. The presence of conformational variants populated even in the absence of the ligand/substrate is a basic concept fitting Darwin’s natural selection and biological evolution.

It may be interesting to quote Henry Buc [5] when he refers to Monod’s approach in writing with Wyman that seminal paper: *“…part of the seduction of the paper rests also on its speculative character. Proteins are considered there not only as transducers working close to their theoretical limit of efficiency, but also as historical objects that evolution has optimized. In its two facets, radiant and convincing, or speculative and questionable, the article bears Monod’s label, an amateur in physical-chemistry, a visionary insofar as molecular Darwinism is concerned”.*

The structure–function relationships in fish hemoglobins have been analyzed within the framework of the MWC allosteric model, keeping in mind that they are all *bona fide* tetramers with general formula α_2_β_2_, containing protoheme as the O_2_ binding site. Fish blood almost always contains several hemoglobins that are functionally distinct. The surprise was that the large functional differences between the principal fish hemoglobins were consistent with the premises of the MWC model, as discussed below. The overall span in O_2_ affinity experienced at different pH values is huge, the p_50_(O_2_) ranging from ~0.5 mmHg for the quaternary R-state to >1000 mmHg for the T-state at pH 6. The large differences in O_2_ binding properties were shown to respond to specific physiological demands of the animal, a paradigmatic example of molecular evolution at work.

## 2. The Swim Bladder

All teleost fish have a gas-containing vesicle helping the animal to keep neutral buoyancy at any depth. The swim bladder is a vital organ associated with the so-called gas gland, an organelle producing lactic acid even in the presence of O_2_ [6]. As the fish dives deeper and deeper, the bladder does not collapse due to the release of O_2_ against an increasing hydrostatic pressure, raising to 50 atm and more. As outlined above, the blood of teleost fish generally contains several hemoglobin components at variance to humans. Some of these hemoglobins are essential in releasing O_2_ into the swim bladder when exposed to the acidic environment produced by the gas gland. The blood flowing through the gas gland in a counter current circulation network called *rete mirabile* responds to acidification with an extreme reduction of hemoglobin O_2_ affinity. This pH-dependent allosteric drop in affinity is essentially an extreme form of the Bohr effect, which was called, after its discoverer, the Root effect [7]. The remarkable feature that attracted attention is that, contrary to human blood, fish blood at pH ~6 is largely deoxygenated even in air (150 mmHg).

In the 1960 and early 1970s, several scientists were curious to test the theory that the presence of a swim bladder implies the presence of a Root effect in the blood. Among others, Austin Riggs and his coworkers [8] were engaged in a project directed to investigate the hemoglobins of fish living at great depths approx. one mile and deeper. Some of the benthic species were captured and investigated [8,9]. Despite some variabilities and rare exceptions, it was confirmed that the presence of a swim bladder is associated with the presence in the blood of Root-effect hemoglobin components. Thus, secretion of O_2_ in the bladder against high or very high pressures (100 atm or more) seems to be the rule, without excluding other mechanisms that may cooperate in activating such a demanding response when living under very high hydrostatic pressures.

The peculiar O_2_ binding properties of fish hemoglobins has solicited over the years the curiosity of many blood physiologists and biochemists. Among others, Rossi Fanelli and Antonini [10] published, in *Nature*, a stimulating paper reporting the biochemical and functional characterizations of tuna fish hemoglobin. These data were later analyzed quite rigorously by Jeffries Wyman [11], who was working at the Biochemistry/Regina Elena Institutes at the University of Rome “La Sapienza”. The peculiar findings on tuna hemoglobin were an incentive to begin years later a long-lasting project on the structure–function properties of the hemoglobins from trout (*Salmo irideus*, now renamed *Oncorhyncus mykiss*). This endeavor began in 1969/70 when professor Antonini was still a member of the Science Faculty at the University of Camerino. Since the beginning, Brunori was involved with Irene Binotti, Giancarlo Falcioni and Bruno Giardina, at the University of Camerino.

A fish hatchery in the nearby village of Fiuminata provided endless quantities of trout’s fresh blood. In agreement with extensive literature on fish blood, we found that the hemolysate contained four hemoglobin components which were identified as trout HbI, HbII, HbIII and HbIV (based on different velocities of migration in the gel). After purification and general characterization [12], it was clear that trout HbIV (almost 60% of the total pigment) was characterized by a Root effect, while both trout HbI and trout HbII (35/40% together) were both characterized by cooperative O_2_ binding but absolutely no Bohr effect. At first, Brunori was surprised and somewhat skeptical, but after proper controls, everybody was convinced, and the project flourished. Around 1972, Earl and Thressa Stadman came to Camerino for a Conference, and after Brunori’s talk on hemoglobin, he was invited to write a review for Current Topics of Cellular Regulation [13].

## 3. Trout HbI, a Bare Cooperative Hemoglobin

Oxygen binding to ferrous trout HbI is clearly cooperative with equilibrium binding parameters somewhat reduced compared to HbA (Figure 1A), the maximum Hill coefficient being n_H_ = 2.0–2.2 compared with n_H_ = 2.8–3.0 for HbA [11,14,15]. CO binding was also shown to be cooperative [16,17].

The shape and affinity of the O_2_ binding isotherm of trout HbI is totally independent of pH and organic phosphates; and it displays a very small temperature dependance [16]. The analysis is simplified because of the absence of αβ subunit non-equivalence and essentially no dissociation into αβ dimers (down to less than µM). The T-to-R switch-over point is just above 2 ([19] and references therein).

The 3D structure of the two allosteric states was solved by Tame and collaborators [20], who crystallized the deoxy and CO derivatives of trout HbI. In human HbA, the salt bridges broken during the allosteric transition [21,22] make an important contribution to the difference in stability between the two allosteric states, but do not seem to be sufficient enough to account for the overall free energy change calculated from the O_2_ binding isotherm, calculated according to Wyman [11].

Briefly, trout HbI is fully consistent with a clean ideal MWC model, with the peculiarity that the ligand-linked R-T conformational equilibrium is insensitive to solvent composition. The relative population of T_0_ and R_0_ is temperature dependent, the enthalpy change for the quaternary transition being ~30 kcal·mol^−1^ of the tetramer [16,17]. Therefore, the population of R_0_ increases with temperature as the value of L_0_ approaches unity at >72 °C. Taking advantage of a remarkable temperature stability, we found that the kinetic behavior followed by flash photolysis is fully consistent with thermodynamics [23]. The lack of heterotropic effects on ligand binding was understood based on the amino acid sequence. The residues proposed by Perutz [21] to be responsible for the Bohr effect and for the DPG effect in trout HbI are either absent or substituted by improper amino acids [24]. This was fully independent convincing evidence in support of Perutz’s hypothesis [21].

### Structural Analysis of Trout HbI

Trout HbI proved to be a paradigmatic case of molecular adaptation to physiological lifesaving demands [13]. Hyperactive fish living in fast running waters and sustaining strenuous muscular exercise (such as salmon) may produce sufficient lactic acid to reduce the blood’s pH and thereby inhibit O_2_ delivery to the tissues if all the hemoglobin has a Root effect. Under physical stress and critical metabolic demands, the presence of a Hb component whose O_2_ affinity and cooperativity are independent of pH fulfills the lifesaving role of an emergency O_2_ supplier. An incredibly consistent and totally independent finding that proved the point was reported by Dennis Powers [25], who observed that fish belonging to the subgenus *Catostomus* living in fast moving waters express a trout HbI-like carrier, whereas members of a different subgenus (*Pantosteus*) living preferentially in pools do not.

A structural interpretation of the behavior of trout HbI is possible due to the resolution of the structures of two derivatives (deoxy and CO) of this hemoglobin by Tame and collaborators [20]. In deoxygenated HbI (PDB ID: 1out), Asp101 on the β subunit makes the connection between the inter-dimer α_1_β_2_ interface and the intra-dimer α_1_β_1_ interface via H-bonds and salt bridges. In HbA, residue 101 is a Glu (see sequence alignment in Figure 2).

The conserved α_1_Asp95 can now make a bond with β_2_Asp101 (3.5 Å), which in turn makes a salt bridge with β_2_Arg104 (3.0 Å). This conserved Arg is linked to α_2_His104 via β_2_Asp108, which is Asn108 in HbA. On CO binding (PDB ID: 1ouu), this relay is not broken, but rearranged so that α_1_Asp95 makes a H-bond with β_2_Tyr41 (Phe41 in HbA). β_2_Asp101 makes a weak electrostatic contact (5 Å) with β_1_Arg104. Moreover, β_2_Arg104 changes conformation, and its guanidinium group makes a H-bond with β_2_Asp108, which in turn breaks its salt bridge with β_1_His104 (from 3.9 to 4.4 Å) (Figure 3).

In deoxy human HbA (PDB ID: 2dn2 [28]), this network cannot exist for two reasons: β_2_Glu101 is too far to interact with α_1_Asp94, and β_2_Arg104 is in a conformation which does not face the α_1_β_2_ interface, while making a H-bond with β_2_Asn139. Furthermore, in HbA-CO (PDB ID: 2dn3 [28]), the contacts are even fewer than in deoxy HbA: α_1_Asp94 makes a H-bond with β_2_Asn102 (2.7 Å), and β_2_Asp99 is in contact with α_1_Thr38 (4.0 Å). β_2_Glu101 and Arg104, which are rotated towards the α_1_β_1_ interface, make a salt bridge between themselves (2.7 Å) and are within 4.0 Å of α_2_Lys99.

Another characteristic of trout HbI, which could explain its cooperativity in the absence of a Bohr effect, is a direct cross-talk between the heme groups in the α and β subunits through the α_1_β_2_ interface, both in deoxygenated and ligand-bound states (Figure 4). There is a symmetric array of aromatic amino acids and conjugated double bonds, through which the electrons can be transferred from one heme to another. In particular, heme vinyls of the α subunits are in contact with αTyr42 ring, in π-cation contact with βArg40, in another cation-π contact with βTyr41 and the β heme vinyl. The distances in each contact are all within 4.0 Å, reaching a minimum of 3.6 Å at the center of the symmetry in the deoxy state (Figure 4). Moreover, the α_2_β_1_ interface is completely symmetrical in both unbound and CO-bound states (Figure 4A vs. Figure 4B), differently from HbA.

In HbA, the network connecting the α_1_ and β_2_ hemes is somewhat similar, being αTyr42 and βArg40 conserved, but in position β41, there is a Phe, an aromatic yet hydrophobic residue, which breaks the symmetry. As for the distances, these residues are evenly spaced in the CO-bound form (3.8–3.7 Å), but in the deoxy tetramer, a change in the βArg40 rotamer (and a switch in the α_1_β_2_ interface) is such that it puts α Tyr42 at almost 6 Å from it and at more than 4.0 Å from the heme vinyl, therefore breaking and blocking any favorable communication.

Furthermore, in trout HbI, the two key residues in the heme pockets different from HbA are Trp46 on the α and Thr91 on the β subunits (Figure 2). They are respectively part of the distal and proximal side of the heme binding pocket, but given the head-to-tail arrangement of the subunits in the tetramer, they lay on the same plane, as can be seen in Figure 4, both pulling the propionates, although in opposite topological directions. Moreover, in trout HbI, in the transition from T_0_ to R_4_, the distal side of the α_1_ subunit rotates about 10°, and at the same time, the α_2_ subunit rotates by about 16°; nevertheless, the distal side remains practically unchanged. At the same time, the proximal side of both β subunits experiences a rotation of 8°, while the distal site stays put. Overall, these concerted movements beautifully maintain the symmetry of the tetramer along the α_1_β_2_ interface.

In HbA, the rotation of the α subunits is just 8° on the proximal and 2° on the distal side, while on the β, it is about 3° on each side of the heme. Hence, the pulling/constraining effect of both substitutions (HbA α Phe46 → HbI Trp46, HbA β Leu91 → HbI Thr91) is quite apparent, inducing a breakage of the inter-dimer symmetry and indirectly proving the involvement of those residues in modulation of ligand affinity.

On top of these quaternary constrains, the series of tertiary contacts and the differences in the inter-dimer interface salt bridges [19] as well as point mutations in residues involved in allosteric effects (Figure 2) make trout HbI an elegant example of heme–heme cooperativity without the need of allosteric effectors. The overall outlook of these molecular features, from primary to quaternary structure, indicates that the thermodynamics of trout HbI result in a change in free energy of the transition from T to R more negative than in the case of human HbA and trout HbIV (see below and [16,17,18]).

## 4. Trout HbIV, a Molecular Transducer

The oxygen binding by ferrous trout HbIV is clearly cooperative at pH around 7.5 with n_H_ = approx. 2.2. However, ligand affinity drops almost 1000-fold as the pH is decreased to ~6, where the binding isotherm tends flatten down around 50% saturation towards a plateau (Figure 1C). The decrease in O_2_ affinity and the progressive loss of cooperativity, as apparent from the figure, was interpreted by Brunori [13] as a progressive stabilization of the T-state upon acidification, from pH 7.4 to 6.0. According to the allosteric theory, L_0_ = ([T_0_]/[R_0_]) increased from approx. 4000 to ~3 million. In 1975, Giardina, Ascoli and Brunori published a paper [29] showing that the optical spectrum in the Soret region of trout HbIV 100% saturated with CO changes with pH from 8.1 to 6.2. The overall amplitude of this difference spectrum and the pK of the titration were both shifted by the addition of 1 mM IHP (inositol hexakisphosphate, a powerful allosteric effector). Importantly, at the same time, Perutz had observed a pH-induced difference spectrum in the near UV (280 to 290 nm) of trout HbIV-CO and stated that this optical perturbation was similar to that reported for liganded human HbA and was attributed to a tryptophan at the α/β subunits interface. Years later, Bellelli and Brunori [30] discovered that the difference spectrum in the Soret region described above is almost identical to that reported by Martino and Ferrone [31] for the T_3_↔R_3_ quaternary transition in human HbA. In summary, there was sufficient experimental evidence to conclude that acidification of trout HbIV-CO leads to a huge stabilization of the low-affinity T-state.

The molecular basis of this extreme heterotropic control called the Root effect [7] is expected to depend on the protonation of a few amino acid side chains that control the T-to-R equilibrium with stabilization of the former quaternary state. One hypothesis presented by Perutz and Brunori [32] was a variation on the classical stereochemical mechanism of Perutz [21], whereby the T-state of HbA is stabilized by salt bridges between (i) the imidazole of the C-terminal βHis146 and βAsp or Glu94 or βGlu90 and (ii) the C-terminal carboxyl of the same βHis146β and αLys40.

In mammals, the SH group of βCys93 forms no H-bonds with its neighbors and takes up one of two alternative positions in the R-state, while in the T-state, βCys93 is out of the Tyr pocket, and it is protected from solvent by βHis146. In fish hemoglobins displaying a Root effect, βCys93 is replaced by Ser, whose side chain in the T-state is engaged in H-bonds with βHis146 and with the peptide NH of the same His, stabilizing the C-terminal salt bridges, increasing the allosteric constant, and decreasing O_2_ affinity. Despite some data on carp Hb, which seemed at the time consistent, this hypothesis was sharply criticized by Yokoyama et al [33]. These authors pointed out that (i) substitution of βCys93 by Ser in HbA did not produce the Root effect and (ii) in the T-state tuna Hb, βAsp94 makes a H-bond with βSer93; nevertheless, the latter residue is too far from the terminal βHis146. The basis for this criticism was possibly inappropriate, since it is known that one single mutation alone rarely recapitulates substantial functional effects.

There is good evidence that low pH stabilizes the T-state, accounting for the decrease in cooperativity and of ligand affinity. However, since at pH 6 the affinity is so low that even at 20 atm of O_2_ only 95% saturation is achieved, the stabilization of the quaternary low affinity state is a necessary, but insufficient condition to account for the Root effect. Therefore, an extreme pH-dependent difference in affinity between the α and β subunits was postulated, and further progress demanded an effort to obtain a complete reliable O_2_-binding isotherm at low pH. At this point, Massimo Coletta joined the “*trout-team*” and tackled the problem from scratch.

After a fairly long struggle, the results published in the Proceedings of the National Academy of Sciences proved crucial [18]. Among other problems, Coletta had to build a high-pressure (up to 24 atm) optical cell to be fitted in the Cary14 spectrophotometer, a reliable sturdy machine with a large enough sample compartment to host the high-pressure *ambaradam* of tubes and valves. Absolutely crucial was the stability of the Hb sample, a difficult problem given that at low pH, oxygenated trout HbIV-O_2_ is pretty much unstable, and the rate of autoxidation is enhanced. Therefore, to limit accumulation of met-Hb, not only the solution had to contain the proper amount of the reducing enzymatic system, but at each pressure, the sample had to be substituted with a fresh aliquot, and the optical spectrum had to be recorded as a function of time to make sure that the solution was properly equilibrated with the gas phase. Finally, to correctly calculate the fractional ligand saturation, each sample had to be made alkaline (pH > 9) to achieve complete saturation. In the end, the results were excellent, and fitting the saturation data helped to move forward in the interpretation of the molecular basis of the Root effect in trout HbIV [18,34].

Analysis of the set of binding curves depicted in Figure 1C was revealing. As alluded above, the saturation curves at the higher pH display heme–heme interactions and reveal a bending of the data in the low and high saturation regimes extrapolating towards the asymptotic T and R-states affinities, as canonical for the MWC model. The isotherm at pH 6.1, where the tetrameric protein is stabilized in the T-state, appears to plateau around 50% saturation and can be fitted with two uncoupled binding curves both consistent with a simple mass law description.

The two successive equilibria at pH 6.1 were tentatively assigned to O_2_ binding to the α and β subunits. This simple hypothesis was confirmed by performing a set of ^13^C NMR experiments in collaboration with the Giacometti brothers, Giovanni and Giorgio, and their coworkers [35]. Binding of ^13^CO to the reduced hemes helped to probe the active site of the α and β subunits. The resonance spectra were informative. The ^13^CO bound to trout HbI displayed a single sharp pH independent resonance at 206.20 ppm, indicating that the interaction between the bound ligand and the neighboring side chains of the α and β subunits must be pretty much the same [35].

On the other hand, in the case of trout HbIV-^13^CO, the NMR spectra showed two well-resolved resonances, and the chemical shift in between was clearly pH dependent. Thus, the data showed unequivocally a heterogeneity in the local environment of the two subunits modulated by pH. Even more interesting was the finding that the intensity of the two resonances, which was the same in the fully saturated trout HbIV-^13^CO, displayed two clearly different intensities when the overall ligand saturation at low pH was reduced to 30%. This was pretty convincing evidence for the validity of the hypothesis outlined above. This work however could not establish which one of the two subunits had the lowest affinity. We had to wait for the crystallographic data by Yokoyama et al. [33] on tuna Hb, obtained for unliganded T-state at pH 5 and 7.5 and for CO-bound R-state at pH 8. In that paper, a clear difference in the heme distal pocket between the α and β subunits was found, uplighting some interesting pH-linked conformational changes unique to fish.

### Structural Analysis of Trout HbIV

The crystal structure of trout HbIV in the oxidized Fe^3+^ state was solved by George Phillips and co-workers [36] at pH 5.7 (PDB ID: 3bom, 1.35 Å resolution, crystallized at 5 °C) and pH 6.3 (PDB ID: 2r1h, 1.9Å resolution, crystallized at 25 °C). The starting protein was purified in the CO-bound state, but since crystallization was carried out in air, the protein in the crystals was oxidized (i.e., Ferric or Met-Hb). The deposited 3D structures at the two acidic pH values have an average r.m.s.d. of 0.3 Å overall, showing no detectable difference in tertiary nor in quaternary structure between pH 5.7 and 6.3. A striking fact, not highlighted by the authors, is that trout Met-HbIV has a T quaternary structure. This is a confirmation of the functional and spectroscopic data already published [18,29,30,34]. To demonstrate that trout Met-HbIV is T, we carried out a superposition of 3bom with deoxy trout HbI (i.e. *T*-state): as expected, the calculated *r.m.s.d.* is 0.95 Å overall and 0.5 Å considering the main chain only. On the other hand, the superposition of Met-HbIV with trout HbI-CO (i.e. R-state) results in an *r.m.s.d*. of 2.37 Å overall. In addition, we carried out a superposition of trout Met-HbIV with human deoxy HbA (2dn2, 1.25 Å resolution, [28]) and found an overall *r.m.s.d*. of 1.44 Å. These structural superpositions are shown in the top panel of Figure 5.

To further substantiate that the quaternary structure of the Met-HbIV derivative is a bona fide T-state, we carried out a superposition of the individual α and β subunits as follows: the αMet-troutHbIV vs. the α trout HbI deoxy and CO-bound; the βMet-troutHbIV vs. deoxy and CO-bound β trout HbI. In all these cases, the overall *r.m.s.d.* is 0.5 Å vs. the deoxy and 0.6 Å vs. the CO-bound. As expected, also vs. deoxy HbA (α or β chains) the *r.m.s.d.* is 0.5 Å (Figure 5 lower panel). Furthermore, in both Met-HbIV structures, we identified: (i) the intra-chain salt bridge between βHis69 and βAsp72, which is known to stabilize the T-state, and (ii) a pyramidal electrostatic interaction across the α_1_β_2_ interface between Asp95 and Thr 97 on α_1_ and Asp101 and Arg104 on β_2_ (Figure 6). In these Met derivatives, two His on the β chains (notably His41 and His69), absent in both trout HbI and in HbA, are close enough to both the α_1_β_2_ interface and the heme pocket. These ionizable residues might contribute to the Root effect by lowering oxygen affinity of trout HbIV.

Analysis of individual interactions (as detailed above) may suggest that this T-state is overstabilized, thereby indicated as “*super T-state*”. We may highlight the role of individual amino acid protonation in lowering the oxygen affinity of trout HbIV as follows: (i) the side chains of βHis41 (Figure 6A) and βHis69 (Figure 6B) are sufficiently close to the α_1_β_2_ interface and to the heme pocket to have an effect on lowering oxygen affinity (both are absent in trout HbI and human HbA, Figure 2); (ii) the intra-chain salt bridge between βHis69 and βAsp72 is known to stabilize the T-state (Figure 6B); (iii) the C-terminal salt bridge of βHis145 with βGlu95, which is not broken (Figure 6C); (iv) an electrostatic interaction across the α_1_β_2_ interface contributed by Asp95 and Thr 97 on α1 and Asp101 and Arg104 on β2.

In neither of the two Met-HbIV structures there is evidence for intra chain perturbation, the four subunits in the asymmetric unit having the same heme environment, with the heme being neither entirely flat nor entirely domed (Figure 5, lower panel). The lack of differences in the tertiary structure among the two subunits of Met-HbIV seems in contrast with the ^13^CO NMR spectra showing two well-resolved pH dependent resonances [35], while these NMR results are consistent with crystallographic data on tuna Hb (1v4x [33]).

If one compares the structures of bluefin tuna Hb [33] and trout HbIV [36], it is apparent that in teleost fish, the extremely low oxygen affinity at acidic pH may be contributed by different routes. In bluefin tuna Hb, the tertiary effects via the destabilization of the H-bond between the distal His and the bound O_2_ might be predominant, while in trout Met-HbIV, overstabilization of the T quaternary structure *via* the α_1_β_2_ interface seems to be important, even though tertiary effects due to point mutations, such as those highlighted in purple in Figure 2, may be important in “sensing” the pH change and transmit the information through the α_1_β_2_ interface (as detailed in Figure 6).

As outlined above, in the T-state trout Met-HbIV, there is no evidence for a structural heterogeneity in the heme binding site of the two subunits, but we cannot exclude that oxidation of the heme iron from ferrous to ferric may change the energetics of the distal cavity. On the other hand, the complementary crystallographic data obtained for the deoxy and CO derivatives of tuna Hb at different pH values [33] were exciting. The quaternary structure of tetrameric tuna fish Hb in the T-state shows that the heme binding site displays at low pH intriguing differences among the α and β subunits. In the crystals of deoxygenated tuna Hb, the heme pocket of the α_2_ subunit shows a concerted motion of His46 and Trp47, whose side chains swing outside the distal pocket and induce a movement towards the solvent of distal His60, which makes a salt bridge with the heme propionate. These large internal motions have no effect on the geometry of the proximal His89, which remains in place while changing from basic to acidic pH [33]. In summary, the large pH-linked conformational change involving distal αHis60 and coupled motions of αTrp47 and αHis46 underlines a tertiary allosteric control mechanism which affects the local interactions between Fe^2+^-bound oxygen and the distal amino acid side chains. This allosterically controlled tertiary structural change may account for the marked functional α/β heterogeneity demonstrated by analysis of the complete oxygen binding isotherm [18] (Figure 1C), although the finding that it seems limited to one subunit only remains puzzling.

## 5. Concluding Remarks

Having a molecular–physiological device to keep neutral buoyancy at any depth is a great advantage to teleost fish compared to cartilaginous creatures that have to swim all the time to control their position. The swim bladder is an effective vesicle demanding a substantial adaptation of the hemoglobin molecule and its oxygen-binding parameters. The pH-driven allosteric control of the major hemoglobin component of trout’s blood, trout HbIV, is a sophisticated mechanism geared at coping with the difficult task of pumping oxygen into the swim bladder against a variable, but generally high, external hydrostatic pressure.

Hemoglobin is a beautiful example of how adaptation to many different external conditions can be achieved by a limited repertoire of molecular motions of key side chains in key positions. Human HbA is the prototype of a sophisticated molecular machine, which maintains cooperative ligand binding over a 100-fold range in absolute affinities and a 1000-fold (or greater) range of the allosteric equilibrium constant. Allosteric control of oxygen affinity involves both quaternary and tertiary conformational changes. In ligand-bound trout HbIV, the stabilization by acid pH of the low-affinity quaternary T-state, which was inferred by optical spectroscopy [29], has been substantiated by crystallography [36]. At the level of the gas gland, the predominance of the low-affinity T-state is a necessary, but not sufficient condition to explain the Root effect. Accurate experiments of oxygen binding at acidic pH (Figure 1C) show a biphasic saturation isotherm consistent with a large difference in affinity between the α and β subunits. ^13^CO-NMR experiments [35] showed clearcut differences in the interactions between the bound ^13^CO and the amino acid side chains coating the ligand binding distal site in the two subunits.

The hypothesis is that the pH-dependent dramatic decrease in oxygen affinity for (at least) one α subunit is the tertiary allosteric effect accounting for oxygen secretion into the swim bladder. It has been convincingly established by kinetics and site-directed mutagenesis of hemoglobin, myoglobin and other heme proteins [38,39,40,41,42,43,44] that oxygen affinity is controlled not only by H-bonding with the distal His(E7), but also by interactions with neighboring amino acid side chains coating the heme binding site (such as position B10 [38,39,40]) and by internal packing defects. We know that by engineering the structure of the distal heme environment, oxygen affinity can be modulated across a huge range of p_50_ from 0.1 to 1000 mmHg. The present description of the trout HbIV allosteric quaternary and tertiary control of oxygen affinity is consistent with, but not identical to, the so-called *small-t/small-r* states introduced by W.A. Eaton and coworkers [45,46] to quantitatively account for the internal dynamics of human HbA triggered by photolysis with fast laser pulses.

In the end, we advance a comment concerning trout HbI, which has an important physiological role just because its cooperative oxygen binding curve is insensitive to the environment and delivery to tissues is regulated exclusively by the O_2_ concentration gradient. L_0_, K_R_ and K_T_ yield a ligand binding curve that is cooperative over a fairly restricted range of O_2_ concentration (Figure 1A). As outlined above, this tetrameric Hb is the perfect case of an allosteric system characterized by cooperative ligand binding, but devoid of heterotropic effects; the binding curve being independent of pH, organic phosphates and temperature, only an effect of Na^+^ on affinity was discovered by Airoldi et al. [47]. Moreover, it displays an unusual temperature stability (up to >70 °C), as well as -tetramer stability. There is no evidence for dissociation into αβ dimers even for the liganded state, nor for heterogeneity in the heme binding site among the two subunits. It was astonishing when amino acid sequence data [24] showed that in trout HbI, all the residues which in human HbA are involved in the allosteric heterotropic effects are either absent or chemically modified, a remarkable demonstration that the theory of M.F. Perutz is correct [21]. The relatively lower O_2_ affinity of trout HbI compared to HbA has been critically discussed by Miele et al. [19]. Based on the work of Szabo and Karplus [22] and Lee and Karplus [48], a comparison of the 3D structure for various derivatives of trout HbI and HbA helped to rationalize the differences in O_2_ affinity and in the energetics of cooperativity. The role of salt bridges and of the α_1_β_2_ interface in the two allosteric states was discussed, and we advise interested people to read Miele et al. [19].

We would like to close by reporting an interesting prophetic consideration from a letter written in October 1959 by Max F. Perutz to Dr. G.R. Pomerat of the Rockefeller Foundation: *‘‘...What is most exciting, perhaps, about our results are their wider implications. If the myoglobin of a whale is like the haemoglobin of a horse, then the structure of these two proteins is probably much the same throughout the animal kingdom. There must be certain standard sequences of amino acids, which all these proteins have in common and which determine the characteristic loops and turns of the chain. These must have developed from a common primeval gene which provided the physiological basis for the development of the higher animals, by making possible the storage and transport of oxygen. Looked at from the point of view of protein chemistry it makes one suspect that proteins are probably grouped into broad classes, and that within each class the main structural features are similar. In other words, one will probably discover a natural history of enzymes which will give one an insight into the biochemical development of the species. All this is for the future…’’* (with permission by Vivien Perutz).

The “future” proved that Perutz’s vision was correct, and due to the open data accumulated in public databases, artificial intelligence algorithms such as ECOD (Evolutionary Classification Of protein Domains [49]), AlphaFold [50] and RoseTTA Fold [51,52] can now predict with accuracy structural features based on the amino acid sequences, while tracing phylogenetic trees and, in the next future, will likely fill in the gap between structure and function.

## Figures and Tables

**Figure 1 biomolecules-13-00572-f001:**
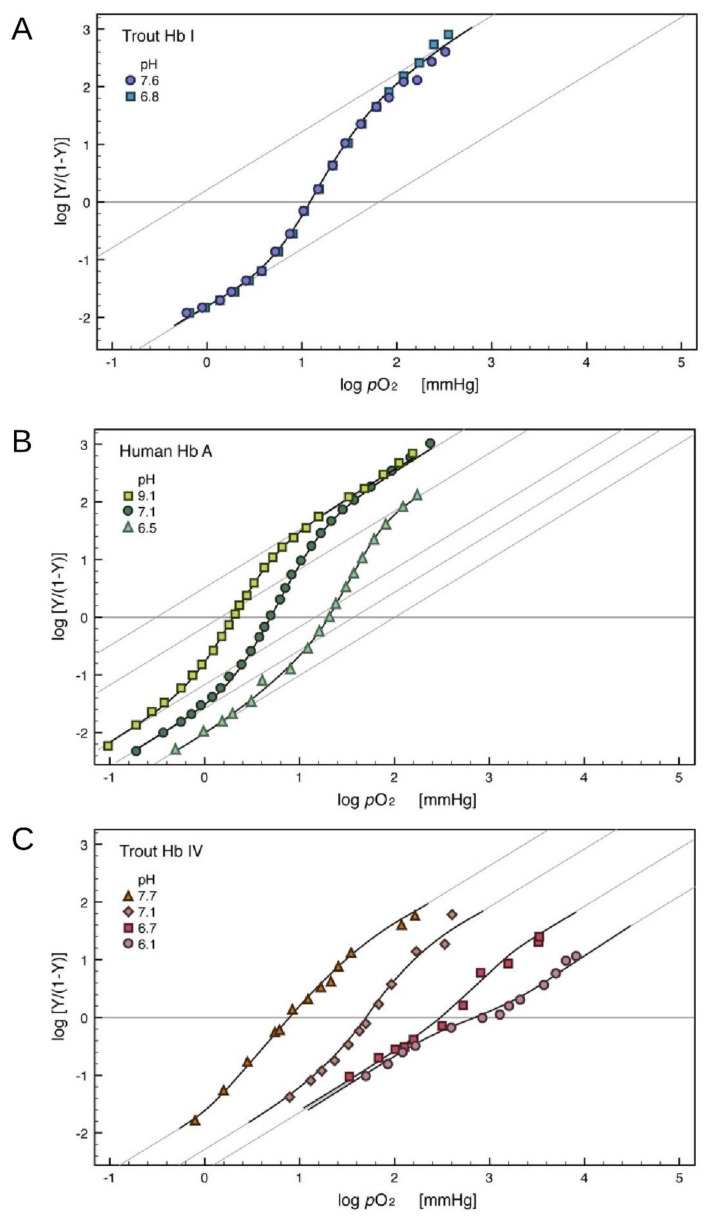
Oxygen equilibrium curves of trout HbI, human HbA, and trout HbIV as a function of pH. Oxygen binding data are plotted according to the Hill equation, where Y is the fractional saturation with the ligand, and pO_2_ is the oxygen partial pressure in the gas phase [11,14]. See also the Appendix A to appreciate the “variation on a theme” *in crescendo*. (**A**) O_2_ binding curves of trout HbI. Data points in blue at two pH values: squares pH 6.8, circles pH 7.6. Solvent 0.2 M bisTris buffer, 4 °C (data from [16]). (**B**) O_2_ binding curves of human HbA. Data points in green showing heterotropic effects. pH values: squares pH 9.1, circles pH 7.4, triangles pH 7.4 plus 2mM Inositol Hexakisphosphate (IHP). Solvent 0.2 M bisTris buffer plus 0.1 M Cl^−^, 20 °C (data from [14,15]). (**C**) O_2_ binding curves of trout HbIV. Data points in red at four different pH values: triangles pH 7.7, diamonds pH 7.1, squares pH 6.7, circles pH 6.1. Solvent 0.05 M in bisTris buffer. A high-pressure optical cell was employed for the very low affinity range, see text (data from [18]).

**Figure 2 biomolecules-13-00572-f002:**
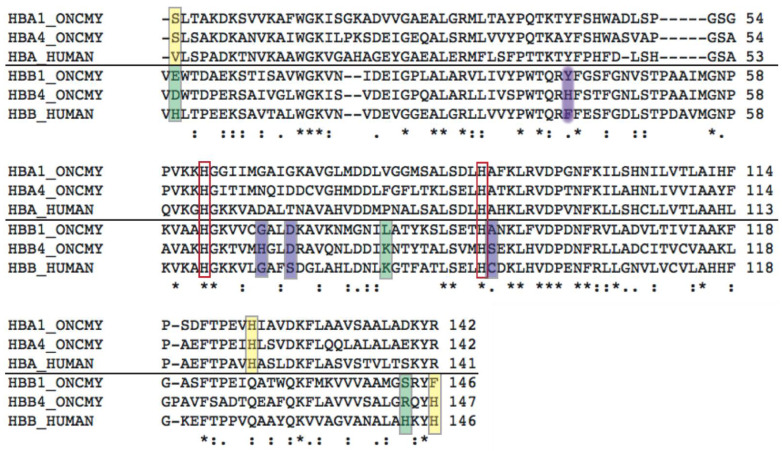
Alignment of the α and β chains of HbA (HBA, HBB) with trout HbI (HBA1, HBB1) and HbIV (HBA4, HBB4). Residues that in HbA are responsible for the Bohr effect (yellow) and DPG binding (green) are mutated in trout-HbI; and those responsible for the Root effect (lilac) in trout HbIV are mutated in HbA. Distal (E7) and proximal His (F8) residues are boxed in red. Acronyms: HBA1_ONCMY, α subunit of trout HbI; HBB1_ONCMY, β subunit of trout HbI; HBA4_ONCMY, α subunit of trout HbIV; HBB4_ONCMY, β subunit of trout HbIV; HBA_HUMAN, α subunit of human HbA; HBB_HUMAN, β subunit of human HbA. The figure was prepared with ClustalOmega [26].

**Figure 3 biomolecules-13-00572-f003:**
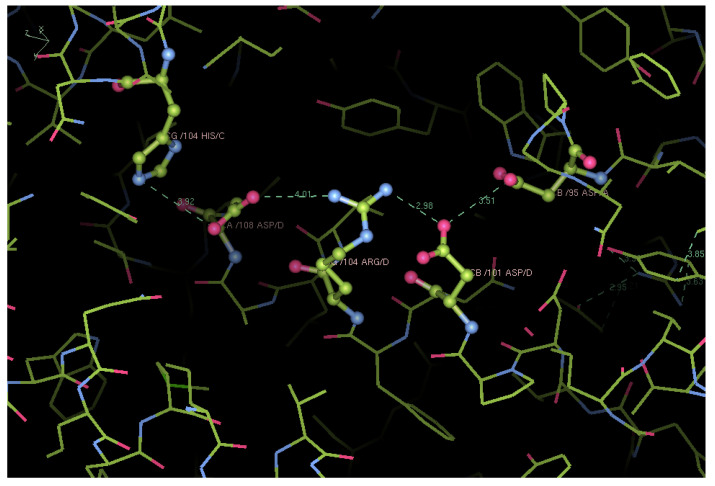
The network of polar contacts linking α_1_β_2_ to α_1_β_1_ interface in deoxy trout HbI. The figure was prepared with COOT [27].

**Figure 4 biomolecules-13-00572-f004:**
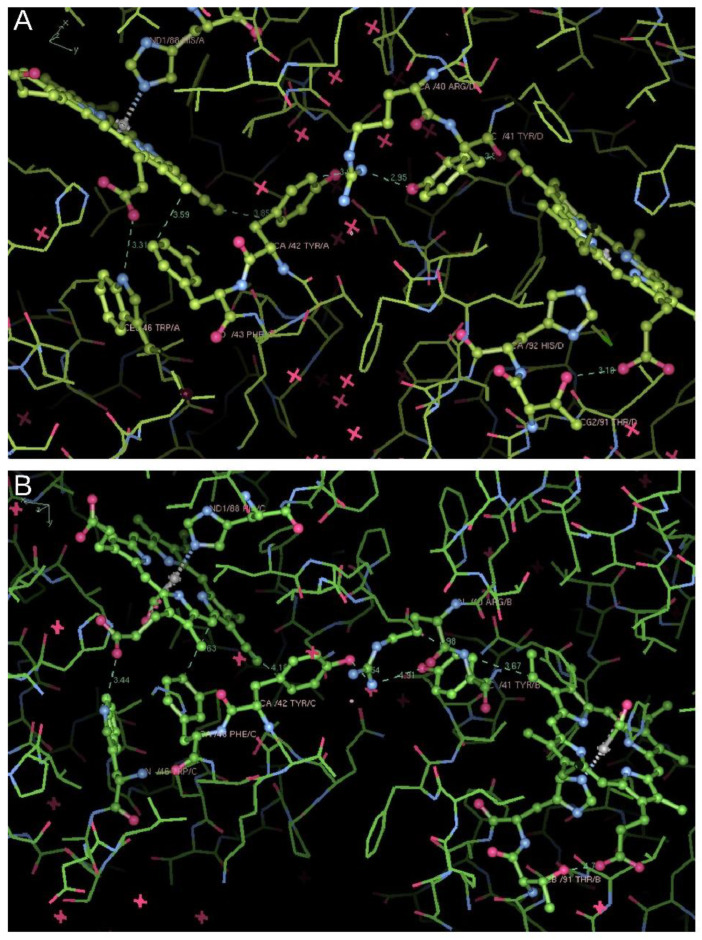
Cross-talk between hemes through the α_1_β_2_ interface in trout HbI: how to connect binding sites more than 15 Å apart. Heme groups and amino acid side chains involved in the cross-talk are in ball and stick representation. (**A**) Heme–heme distances in deoxygenated trout HbI (dashed lines): heme α vinyl—−4.0 Å—Tyr42—−3.6 Å—Arg40—4.0 Å—Tyr42—3.7 Å—heme β vinyl. (**B**) Heme–heme distances in trout HbI-CO: heme α vinyl—−3.7 Å—Tyr42—−3.6 Å—Arg40—3.8 Å—Tyr42—3.8 Å—heme β vinyl. The figure was prepared with COOT [27].

**Figure 5 biomolecules-13-00572-f005:**
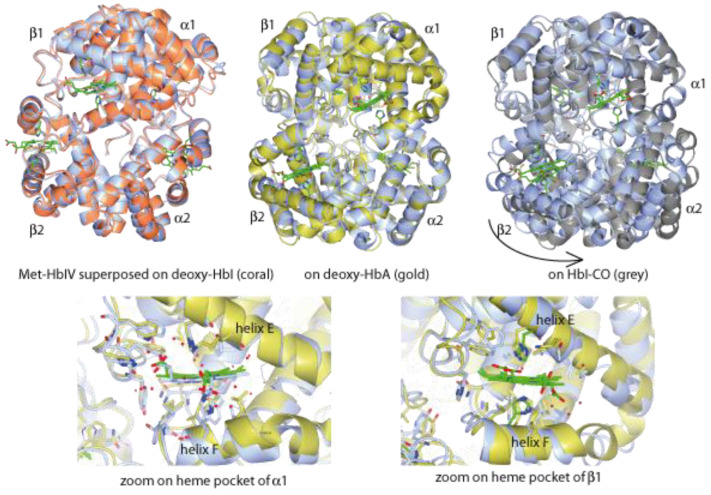
Top row. Structural superposition of trout and human hemoglobins. Ribbon representation of trout Met-HbIV (3bom) in light blue superimposed on: trout HbI deoxygenated (1out, coral, left panel); human HbA deoxygenated (2dn2, gold, centra panel); trout HbI CO-bound (1ouu, grey, right panel). The black arrow highlights the concerted movements. Bottom row. Zoom on the α and β heme pockets of superimposed trout Met-HbIV and deoxy HbA. In stick representation are the amino acids lining the distal and proximal side of the heme pockets, as well as the ferric vs. ferrous unbound hemes. The figure was prepared with CCP4mg [37].

**Figure 6 biomolecules-13-00572-f006:**
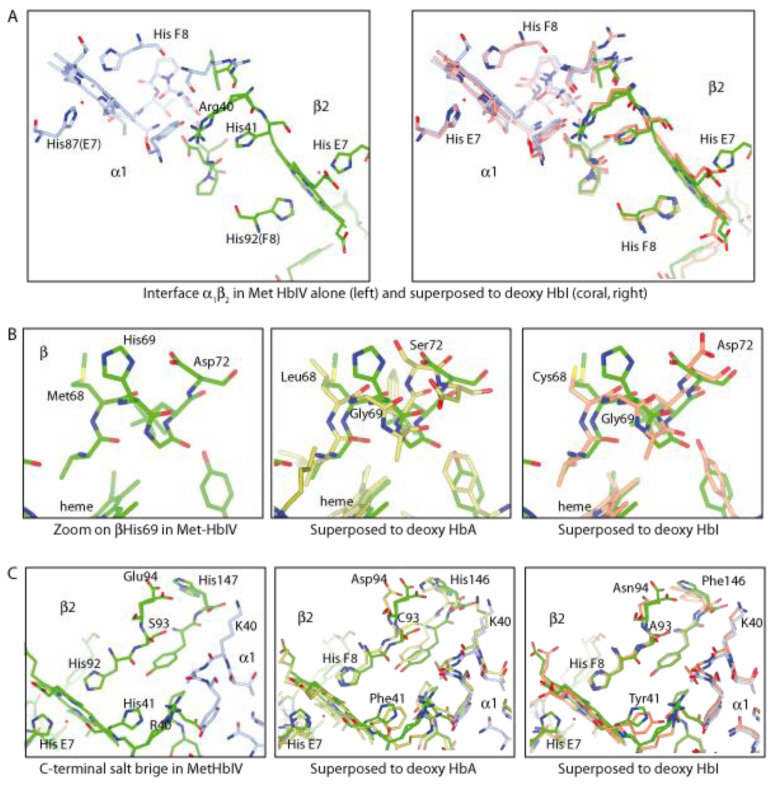
Zoom over the inter-dimer interface in trout Met-HbIV (β chain in green and α chain in light blue) and residues responsible for lowering affinity by exerting the Root effect. (**A**) Cross-talk between the heme pockets through the α_1_β_2_ interface in trout Met-HbIV (left) as compared and superimposed to deoxy trout HbI (coral); βHis 41 in Met-HbIV is mutated into Tyr in HbI, which will not be able to sense the pH change and transmit it through the inter-dimer interface. (**B**) Surroundings of βHis69 on the edge of the heme vinyl group in Met-HbIV (left frame), in comparison with deoxy HbA (gold, middle frame) where His is a Gly and Asp72 is a Ser, and in comparison with trout HbI deoxy (coral), where Asp is conserved, but His is substituted by a Gly as in the human protein. (**C**) Zoom on the C-terminal salt bridge (β His147-Glu94) of Met-HbIV (left frame), superimposed to deoxy HbA (gold, middle frame) and deoxy trout HbI (coral, right frame). As a reminder, βGlu94 is on the same F helix holding the proximal His(F8), which coordinates the heme iron. The figure was prepared with CCP4mg [37].

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
