# Peer review of "Modulation of Allosteric Control and Evolution of Hemoglobin"

_biomolecules, 2023, doi:10.3390/biom13030572_

Round 1

Reviewer 1 Report

The manuscript reviews several decades of research regarding the allosteric control of oxygen binding by hemoglobin, starting back at the Alpha Helix expedition almost 50 years ago. Following this very interesting historical preamble, the review focuses on the comparison of the hemoglobins from various fish species. This group of hemoglobins if of particular interest, as they can display a remarkably large span in their oxygen affinity for different pH values, which appears to be a response to the animal physiological needs, as hemoglobins can be led to fulfill various biological functions.

The thorough analysis of the structure of various hemoglobins highlights the key role played by the interaction network formed by the residues located at the interface between each globin subunit in the tetrameric assembly. Additional inclusion of evolutionary data shows how some of these interactions can be disrupted by specific point mutations, thus leading to a change in the hemoglobin activity from one specie to the other.

The review is very interesting and well written and I only have one minor comment for improvement :

- As the structural analysis repeatedly highlights the importance of residues Asp101 and Asp104 in the inter-and intra-dimer interfaces, it might be worth mentioning the work of Stadler et al. (https://doi.org/10.1098/rsif.2012.0364) which showed how these particular residues present specific mechanical properties that are extremely sensitive to small conformational changes within the protein.

Author Response

We thank the reviewer for their very kind comments.

"As the structural analysis repeatedly highlights the importance of residues Asp101 and Asp104 in the inter-and intra-dimer interfaces, it might be worth mentioning the work of Stadler et al. (https://doi.org/10.1098/rsif.2012.0364) which showed how these particular residues present specific mechanical properties that are extremely sensitive to small conformational changes within the protein."

We have read the suggested paper, which is a very interesting piece of comparative work on vertebrate hemoglobins. However, we think that its citation in this review is not pertinent, since we do not infer any thermal stability, nor we discuss any mechanism of folding.

Reviewer 2 Report

The manuscript by Brunori and Miele reports on the functional and regulatory properties of fish hemoglobin that the Rome’s group deeply investigated in the period 1960-1980. The review, clearly written, describes some of the key feature of fish hemoglobins, the most peculiar being the extreme Bohr effect, named Root effect. Authors correlate functional and regulatory properties with structural data to gain insight into allosteric mechanisms that are discussed within the frame of the MWC model.

I have only a very minor point. I suggest to substitute the term “secrete” for the unloading of oxygen from hemoglobin with “release”, for example “Some of these hemoglobins are essential to secrete O2 …”(page 2, section 2).

Author Response

Thank you very much for your kind comments. We are glad to have shown the structure-function relationships in a clear manner.

"I have only a very minor point. I suggest to substitute the term “secrete” for the unloading of oxygen from hemoglobin with “release”, for example “Some of these hemoglobins are essential to secrete O2 …”(page 2, section 2)."

We have modified accordingly.

Reviewer 3 Report

This is a well written and an elegant review article on modulation of allosteric control and evolution of hemoglobin. This article is of  paramount importance; contributing enormously to our understanding of evolution of allostery in hemoglobin.  Please address the few comments below prior to publication.

1.      Page 6, last paragraph: I wouldn’t consider the 5Å contact between β2Asp101 and β1Arg104 as a weak salt bridge. At this distance, there is clearly no interaction between the two residues. Same as the supposedly 3.9 Å salt bridge interaction between  β2Arg104 and β2Asp108.  

2.      Page 9: Reword the following sentence to read better “The decrease in O2 affinity and the progressive loss of cooperativity as apparent from the figure where interpreted by Brunori within the framework of the allosteric theory as a progressive stabilization of the T-state by acidification from pH 7.4 to pH 6.0, with L0=([T0]/[R0]) increasing from approx 4.000 to ~3 million.”

3.      Page 10: Reword the following sentences to read better: “The 13CO bound to the reduced hemes allowed to probe the active site of the α and β subunits.”

Author Response

We are grateful to the reviewer for their very constructive comments, which we have addressed to our best.

1. Page 6, last paragraph: I wouldn’t consider the 5Å contact between β2Asp101 and β1Arg104 as a weak salt bridge. At this distance, there is clearly no interaction between the two residues. Same as the supposedly 3.9 Å salt bridge interaction between β2Arg104 and β2Asp108.

We have modified the sentence as follows: "β2Asp101 makes a very weak electrostatic contact (5 Å) with β1Arg104. Moreover, β2Arg104 changes conformation and its guanidinium group makes a H-bond with β2Asp108, which in turn breaks its salt bridge with β1His104 (from 3.9 to 4.4 Å) (Figure 3)."

2. Page 9: Reword the following sentence to read better “The decrease in O2 affinity and the progressive loss of cooperativity as apparent from the figure where interpreted by Brunori within the framework of the allosteric theory as a progressive stabilization of the T-state by acidification from pH 7.4 to pH 6.0, with L0=([T0]/[R0]) increasing from approx 4.000 to ~3 million.”

Thanks, we have modified the sentence as follows, to increase readability: "The decrease in O2 affinity and the progressive loss of cooperativity, as apparent from the figure, where interpreted by Brunori [13] as a progressive stabilization of the T-state upon acidification, from pH 7.4 to pH 6.0. According to the allosteric theory, L0=([T0]/[R0]) increased from approx 4,000 to ~3 million."

3. Page 10: Reword the following sentences to read better: “The 13CO bound to the reduced hemes allowed to probe the active site of the α and β subunits.

We thank for this remark and we have changed as follows: "Binding of 13CO to the reduced hemes allowed to probe the active site of the α and β subunits."

Reviewer 4 Report

The concept of explaining the function of a protein in terms of its detailed atomic structure began with the famous paper of Max Perutz on his stereochemical mechanism for the function of hemoglobin.  This work together with the allosteric model of Monod, Wyman and Changeux (MWC) sparked an enormous amount of research that continues to this day.  The Rome group led by Eraldo Antonini and Maurizio Brunori contributed greatly in this area by careful equilibrium and kinetic experiments on fish hemoglobins, especially the 4 components of hemoglobin found in trout.  The studies of all 4 trout hemoglobin provided convincing evidence to support both the MWC model and Perutz’s assignment of residues responsible for the Bohr effect and Root effects – the effect of pH on affinity – a key part of his mechanism. One of the most striking observations was the enormous (~1,000-fold) change in affinity of trout IV hemoglobin, which is central to understanding the physiology of the swim bladder

In this work, Brunori and Miela provide an updated account of the role of this work on understanding structure-function relations in hemoglobin, taking advantage of the numerous X-ray structures that have been solved in recent years to solidify their conclusions.  It is an informative, scholarly, and elegant presentation of this important research subject, which comes as no surprise since Brunori has been the leading intellectual in the hemoglobin field for over 50 years.  I strongly recommend it for publication without any changes (a couple of typos – “revolution” instead of “evolution” and “1,000” or “1000” instead of “1.000.”)

Author Response

We would like to thank a lot this reviewer, who has really understood our point on the importance of bringing together and recollecting all known functional and structural data to highlight the ensemble vision on a complex system.

"In this work, Brunori and Miele provide an updated account of the role of this work on understanding structure-function relations in hemoglobin, taking advantage of the numerous X-ray structures that have been solved in recent years to solidify their conclusions. It is an informative, scholarly, and elegant presentation of this important research subject, which comes as no surprise since Brunori has been the leading intellectual in the hemoglobin field for over 50 years. I strongly recommend it for publication without any changes (a couple of typos – “revolution” instead of “evolution” and “1,000” or “1000” instead of “1.000.”)"

We have modified accordingly all the typos.